# Comparison of Different Machine Learning Methods for Predicting Cation Exchange Capacity Using Environmental and Remote Sensing Data

**DOI:** 10.3390/s22186890

**Published:** 2022-09-13

**Authors:** Sanaz Saidi, Shamsollah Ayoubi, Mehran Shirvani, Kamran Azizi, Mojtaba Zeraatpisheh

**Affiliations:** 1Department of Soil Science, College of Agriculture, Isfahan University of Technology, Isfahan 8415683111, Iran; 2Rubenstein School of Environment and Natural Resources, University of Vermont, 81 Carrigan Drive, Burlington, VT 05405, USA; 3Gund Institute for Environment, University of Vermont, 210 Colchester Ave, Burlington, VT 05401, USA

**Keywords:** clay type, mineralogy, remote sensing indices, valley depth, soil modeling, machine learning

## Abstract

This study was conducted to examine the capability of topographic features and remote sensing data in combination with other auxiliary environmental variables (geology and geomorphology) to predict CEC by using different machine learning models ((random forest (RF), k-nearest neighbors (kNNs), Cubist model (Cu), and support vector machines (SVMs)) in the west of Iran. Accordingly, the collection of ninety-seven soil samples was performed from the surface layer (0–20 cm), and a number of soil properties and X-ray analyses, as well as CEC, were determined in the laboratory. The X-ray analysis showed that the clay types as the main dominant factor on CEC varied from illite to smectite. The results of modeling also displayed that in the training dataset based on 10-fold cross-validation, RF was identified as the best model for predicting CEC (R^2^ = 0.86; root mean square error: RMSE = 2.76; ratio of performance to deviation: RPD = 2.67), whereas the Cu model outperformed in the validation dataset (R^2^ = 0.49; RMSE = 4.51; RPD = 1.43)). RF, the best and most accurate model, was thus used to prepare the CEC map. The results confirm higher CEC in the early Quaternary deposits along with higher soil development and enrichment with smectite and vermiculite. On the other hand, lower CEC was observed in mountainous and coarse-textured soils (silt loam and sandy loam). The important variable analysis also showed that some topographic attributes (valley depth, elevation, slope, terrain ruggedness index—TRI) and remotely sensed data (ferric oxides, normalized difference moisture index—NDMI, and salinity index) could be considered as the most imperative variables explaining the variability of CEC by the best model in the study area.

## 1. Introduction

The soil cation exchange capacity (CEC) is the summation of exchangeable cations (e.g., magnesium, calcium, sodium, and potassium) that could be adsorbed at a definite pH [1]. CEC serves a noteworthy function in adsorbing and releasing nutrients required by plants and evaluating the potential risk of inorganic and some cationic organic pollutants [2]. Moreover, as a global challenge soil, CEC is important in predicting the occurrence of carbon sequestration mechanisms [3]. In addition, CEC has been identified as an indicator of structural stability and soil shrinkage–swelling behavior in vertisols [4]. Clay content, clay types, and soil organic matter have important contributions to controlling soil CEC owing to their considerable specific surface area as well as predominantly negative charges [5]. Soil reaction in pH-dependent soils can also significantly affect the CEC value. Clay types have an important contribution to mediating the CEC in soils. Kaolinite and montmorillonite are the extreme types of clays that might lead to low and high CEC in soils [6].

There are several laboratory methods for the measurement of CEC. The most common method, as described in [7], is to wash the soil with sodium acetate, remove excess soluble salts with ethyl alcohol, and measure the sodium obtained following ammonium rinsing acetate. Nevertheless, the laboratory methods are time-consuming and expensive, so they need several leaching processes. Moreover, the laboratory measurement of CEC in Aridisols of Iran is faced with more challenges because of the presence of some nonclay minerals, such as calcium carbonate and gypsum [8].

In recent years, to overcome these deficits, as mentioned earlier, several pedotransfer functions (PTFs) have been established worldwide to estimate CEC using easily accessible soil data. The most priority method was developing PTFs by using easily available soil properties. In this regard, some scholars developed multiple linear regression models to predict CEC [9]. Then, researchers employed intelligent systems, such as artificial neural networks (ANNs) to determine the nonlinear relationships between soil data and CEC [10]. Asadzadeh et al. [11], in a study in calcareous soils of East Azerbaijan Province, northwest Iran, for developing PTFs by multiple linear regression and artificial neural network, reported no significant difference between the two methods based upon the performance criteria. Furthermore, the simultaneous spatial prediction was developed initially using classical geostatistical approaches [12,13]. Meanwhile, some researchers have claimed that using geostatistical methods involves disadvantages for spatially predicting soil properties, in addition to soil CEC; the major deficit is requiring too many sampling numbers and the necessity to have high spatial dependency [14,15].

Recently, digital soil mapping (DSM) has been employed to incorporate easily available data to predict expensive and labor-consuming soil properties. Various geospatial data, such as remotely sensed data, proximal sensing data attributes, which are obtained from the digital elevation model (DEM), and legacy soil data could be applied for predicting target properties in DSM. A range of machine learning approaches has been employed, from linear statistical to intelligent models (e.g., [9,16]).

Regarding the spatial prediction of CEC, Triantafilis et al. [4,17] used electromagnetic data derived from EM38 ECa for mapping CEC at the field level [17]. Taghizadeh et al. [18] also applied DEM and LANDSAT data for mapping CEC at multiple depths [18]. Huang et al. [1] used an error budget method to compute the different errors of a CEC digital map by utilizing gamma-ray spectroscopy and the data related to apparent electrical conductivity (ECa), showing that individual covariate errors by the gamma-ray and ECa error were large [1]. Nussbaum et al. [19] used a variety of auxiliary variables and predictive models to predict effective CEC (eCEC), indicating that the tree-based method boosted regression tree (BRT) and, in particular, random forest (RF) outperformed, on average [19]. Sorenson et al. [20] also predicted soil properties and CEC in Saskatchewan, Canada, using bare soil composite imagery obtained from multitemporal satellite imagery. They found the predictive models, which the RF model developed, had an R^2^ equal to 0.50, with the RMSE being 5.7 meq/100 g [20].

Although several research studies have investigated soil property variability by using DSM, little attempts have been made to address CEC variability in arid and semiarid regions by combining various auxiliary variables, which is necessary for management and precision agriculture. Therefore, this research study aimed to (i) predict CEC by combining various environmental variables, remote sensing indices, and thematic maps; (ii) compare the capability of some different machine learning methods; and (iii) determine the important variables for the explanation of the CEC variability in the given study area in a semiarid region in the west of Iran.

## 2. Materials and Methods

### 2.1. Study Area and Soil Sampling

The current research study was implemented in Kurdistan province, western Iran, which is between the latitudes 35°05′00″ and 35°20′00″ N and the longitudes 47°12′59″ and 47°39′17″ E (Figure 1). This area is approximately 110,000 ha, with the mean altitude being 2277 m above sea level. This area has a mean annual temperature of 10.20 °C as well as a mean annual rainfall of 370 mm. The rain mostly falls in spring and winter. In addition, this area has the soil moisture and temperature regimes of Xeric and Mesic, respectively [21]. The area’s lithological setting comprises reddish sandy marls and marl sandstone, river deposits, alluvium-cultivated land, and granite [22]. Soil sampling was carried out by applying the stratified random sampling approach. Ninety-seven samples were obtained from the studied area’s surface layers (0–20 cm depth) (Figure 1). The coordinates of each sample were recorded by using GPS.

The soil cation exchange capacity (CEC) was measured by saturation with potassium and then replaced with saturating ammonium cation, as described by Rhoades [7]. Moreover, soil properties, such as soil particle size distribution, were checked by the Pipette method [23]. Soil organic carbon was measured by wet oxidation [24] in all 97 soil samples. Kittrick and Hope’s (1963) method separated the clay fraction from the bulk soil in four soil samples with different clay activities. An analysis of the clay-oriented samples of Mg-saturated, ethylene glycol (EG)-solvated, and K-saturated at 25, 330, and 550 °C was performed by X-ray diffraction (XRD) by applying a D8 ADVANCE diffractometer with CuKα radiation (40 kV, 40 mA). The peak area related to the 001 reflections for the considered main clay minerals (smectite = 17 Å, illite = 10 Å, and kaolinite/chlorite = 7 Å) on the EG-treated sample could be applied for the semiquantitative calculation of the clay minerals [25].

### 2.2. Environmental Covariates

This study used three sources of environmental variables to extract the auxiliary variables to predict CEC; these included (i) topographic attributes derived from DEM, (ii) remotely sensed data, (iii) and thematic maps, such as geology and geomorphology maps (see Table 1 for detailed information).

DEM was obtained from the Shuttle Radar Topography Mission (SRTM) [26], with a spatial resolution of 30 m; it was applied to obtain 14 topographic attributes by applying SAGA GIS [26]. Such topographic attributes are often utilized in DSM studies for the representation of variability in topography (Table 1). The imagery related to the Landsat 8 Data Continuity Mission (Landsat 8) sensor that had been previously corrected in an atmospheric manner and was applied to prepare the covariates from remote sensing data.

Remote sensing data were obtained from Landsat 8 (Operational Land Imager; OLI) on 7 August 2020. Projection of Landsat image (30-m spatial resolution) was made by applying the WGS 1984/UTM zone 38N map projection. Following corrections made on the Landsat 8 images, 23 covariates were subjected to calculation. These covariates were the original Landsat bands, vegetative indices (NDVI, SAVI, TDVI, NDMI, and NDSI), soil salinity indices, clay mineral index, and various iron oxide indices, as well as bare soil index (BI) (see Table 1).

Thematic maps included geomorphology and geology maps. Based on the geology map, the studied area’s lithological setting comprised reddish sandy marls and marl sandstone (Pliocene) with higher quantities of calcium carbonate, river deposits (Quaternary), alluvium-cultivated land (Quaternary), granite, and granodiorite (Pre-Cretaceous, more likely Upper Jurassic) [22]. The Cretaceous formations included diverse metamorphic rocks, in addition to igneous rocks (Figure 2). An illustration of the distribution of geologic and geomorphic units is presented in Figure 2. All raster-based covariates (aggregation or disaggregation) were resampled to a 30 m conjoint spatial resolution.

### 2.3. Preprocessing for Environmental Variables and Feature Selection

When there is a large set of predictors, enhancement of a model’s performance and generalization capability needs exhaustive exploration of all subsets possible for selecting the best predictors [27]. Use was made of the Boruta method for the identification of statistically “all-relevant” environmental covariates. Secondly, use was made of multicollinearity analysis for the removal of highly correlated variables from the modeling process in order to enhance the prediction’s performance [28]. The Boruta algorithm can detect linear and nonlinear relationships among CEC, clay activity, and environmental covariates as it is based on a random forest (RF) classification algorithm [29]. Therefore, the Boruta algorithm can provide five random probes acquired by shuffling the environmental covariate values to reduce their collinearity with dependent variables (i.e., CEC and clay activity). Then, RF regression is implemented on the combination of the environmental variable and random probes; each variable’s importance can determine the Z score. Subsequently, identification of the maximum Z score is performed among the random probes (MZRPs); this can act as a reference to find if an environmental variable is relevant to the activity of CEC and clay with a two-sided test of equality. The environmental covariates with Z scores were considered the relevant variable significantly more than MZRPs [30]. A complete account of the Boruta algorithm can be seen in Xiong et al. (2014). The Boruta package [30] was applied to implement the Boruta “all-relevant” searching method in the R statistical software [31].

### 2.4. Modeling Approaches

As mentioned above, in this study, three categories of variables were used as the input variables for predicting CEC and clay activity. In particular, we examined four techniques; these included k-nearest neighbors (kNNs), support vector machines (SVMs), random forest (RF), and Cubist (Cu) models. The k-nearest neighbor (kNN) algorithm is regarded as a nonparametric learning cornerstone. Due to its simplicity and flexibility, such procedures are selected in many scenarios, especially in settings where a complex underlying model is the case (for more details, see [32]). Support vector machines (SVMs) are regarded as state-of-the-art neural network technology, which is based on statistical learning [33]. SVMs use a linear model to implement nonlinear class boundaries through the input vector’s nonlinear mapping into a high-dimensional feature space. The linear model, which is developed in the novel space, can display a nonlinear decision boundary in the original space (see details in [33]). Random forest (RF) is considered a classifier or regression model consisting of many decision or regression trees. Every tree is dependent on the values of a random vector sampled independently and with one single distribution for all trees in the data (see details in [34]). Cubist is considered as a Quinlan’s M5 model tree extension [35], just like common regression trees, with the exception that the leaves are some sort of linear regression of the covariates. In Cubist, the prediction is on the basis of linear regression models rather than discrete values (see details in [36]). The implementation of feature selection and modeling was performed done by applying the “*Boruta*” and “*caret*” packages in R 3.5.2 [31], respectively.

### 2.5. Assessment of Models

A ten-fold cross-validation approach involving ten replications was applied to evaluate the four models’ prediction performance [37]. All of the models were assessed based on the common performance metrics: mean absolute error (MAE), root mean square error (RMSE), coefficient of determination (R^2^), squared correlation coefficient (r^2^), Lin’s concordance coefficient (rhoC), ratio of performance to deviation (RPD), and ratio of performance to interquartile distance (RPIQ).
(1)$$MAE=\frac{1}{N}\;\sum_{i=1}^{N}[\;\left|Z\left(X_{i}\right)-Z^{*}\left(X_{i}\right)\right|]$$
(2)$$RMSE={\left[\frac{\sum_{i=1}^{N}{\left[Z\left(X_{i}\right)-Z^{*}\left(X_{i}\right)\right]}^{2}}{N}\right]}^{1/2}$$
(3)$$RPD=\frac{\sqrt{\sum_{i=1}^{n}{\left(Z\left(X_{i}\right)-Z\bar{\left(X\right)}\right)}^{2}}}{\sqrt{\sum_{i=1}^{n}{\left(Z^{*}\left(X_{i}\right)-Z\left(X_{i}\right)\right)}^{2}}}$$
(4)$$\mathrm{RPIQ}=\frac{\left(Q_{3}\left(obs\right)-Q_{1}\left(obs\right)\right)}{\sqrt{\frac{1}{n}\sum_{i=1}^{n}\sum_{i=1}^{n}{\left(obs_{i}-\bar{pred}\right)}^{2}}}$$

In the above, Z^∗^(X_i_) and Z(X_i_) refer to the predicted and observed values, respectively; N stands for the number of the measurement; $\bar{Z}\left(X\right)$ denotes the observed value average; µ_obs_ and µ_pred_ indicate the means of observed and predicted values; $\partial_{obs}^{2}$ and $\partial_{pred}^{2}$ represent the corresponding variance; *r* signifies Pearson’s correlation coefficient between observed and predicted values, Q1(obs) and Q3(obs) are regarded as the first (25%) and third (75%) quantiles of the observations, and Q_3_(obs)−Q_1_(obs) displays the interquartile distance. According to validation metrics, lower ME and RMSE and higher R^2^, CCC, RPD, and RPIQ values can indicate a more desirable model performance and correct calculation with less error. RMSE and MAE can make the diagnosis of the error variations in predictions. Moreover, RPD displays the relative improvement in the modeling process; an RPD value beyond one shows the model’s improvement [38].

## 3. Results and Discussion

### 3.1. Variability of CEC, Soil Properties, and XRD Analyses

Descriptive statistics of CEC show that this variable ranged from 11.15 to 50.83 cmole_c_ kg^−1,^ with a range of 39.68 cmole_c_ kg^−1^. The skewness value of 0.60 confirmed the normal distribution of CEC in the study area (Table 2). Soil organic carbon (SOC) varied from 0.33 to 6.63%, with a high CV value (CV = 68.75%) because of the high variation in land uses and management practices. pH values varied from 6.72 to 7.78 with a mean of 7.32 and low variation (CV = 3%). The studied soils are not saline, and the mean of EC is about 0.23 dS m^−1^ due to relatively high precipitation related to arid regions.

The high variation of soil CEC (CV = 27.26%) in the given area (Table 2) could be mainly attributed to various contributions of clays with different activities in different locations. According to the results of the XRD analysis, the most active soil in the studied area with CEC = 38.7 cmole_c_ kg^−1^ and clay activity (CA = 2.85) comprised high relative contents of vermiculite (d-space = 14 °A with MgCl_2_ and 15.5 °A, following ethylene glycol treatment and a complete collapse after KCl treatment to 10 °A) (see Figure 3a). Vermiculite clay normally has a CEC between 130 and 210 cmole_c_ kg^−1^, with a significant contribution to leading to high CEC. Other important clays with a lower contribution in this soil were illite and kaolinite. In the second group of soils in the area with CEC around 26 cmole_c_ kg^−1^ and CA = 1.5, the predominant clays were vermiculite and montmorillonite (Peak 17 °A following EG treatment), chlorite, illite, and kaolinite (Figure 3b). It is well known that montmorillonite has lower CEC than vermiculite, around 70–1520 cmole_c_ kg^−1^ [39]. In the third group with the lowest CEC and CA, 18.3 cmole_c_ kg^−1^ and 1.02, respectively, lower vermiculite was observed with no evidence of montmorillonite and the higher contribution of chlorite, illite, and kaolinite (Figure 3c). Obviously, kaolinite (1–15 cmole_c_ kg^−1^), illite (10–40 cmole_c_ kg^−1^), and chlorite (10–40 cmole_c_ kg^−1^) have a low cation exchange capacity when compared to montmorillonite and vermiculite [39].

### 3.2. Modeling and Spatial Prediction of CEC

The present study used three input auxiliary variables and cation exchangeable capacity (CEC) as three machine learning methods and modeled the target variable, which included kNN, SVM, RF, and Cu models. In the first stage, to find the most important variables that could control the variability of CEC, feature selection was performed by the Brouta approach. Given input variables, some methods had to be developed to minimize and optimize the input attributes or feature selection process [40]. As presented in Figure 4, the feature selection results show that among the topographic attributes, valley depth, elevation, slope, terrain ruggedness index (TRI), MRVBF (Multiresolution valley bottom flatness index), and effective airflow heights were the most important ones. In regard to the remote sensing indices, the ferrous minerals index, Band 1, Band 3, gossan, ferric oxide index, normalized difference moisture index (NDMI), salinity index, and shortwave infrared (SWIR) were the most imperative. Among the top twenty important variables, geology as the thematic map was identified, and geomorphology did not make any contribution.

The most important parameters of the studied models for the prediction of CEC in the considered study area by the four selected models are presented in Table 3. For the kNN model, the best-fitted k was obtained at 19. For the RF model, ntree and mtry were obtained at 550 and 5, respectively; regarding the SVM model, sigma and C were obtained at 0.3744 and 0.5, respectively. Finally, for the Cu model, committees and neighbors were 10 and 0, at the best run, respectively.

The average validation criteria for cation exchange capacity (CEC) prediction in the studied area are represented in Table 4. Among the employed models in the training dataset, the most and least accuracy was obtained by the RF and kNN models for predicting soil CEC. Among the four selected models, the following performance was achieved in ranking: RF > SVM > Cu > kNN. In this stage, RF with R^2^ = 0.86, RMSE = 2.74 cmole_c_ kg^−1^, and RPIQ = 3.74 had the best performance. Contrary to the validation results, the most precise and accurate model in the validation dataset was the Cu one with R^2^ = 0.49, RMSE = 4.51 cmole_c_ kg^−1^, and RPIQ = 1.71. In this stage, similar to the training stage, kNN showed the least performance with R^2^ = 0.19, RMSE = 5.72 cmole_c_ kg^−1^, and RPIQ = 1.34. The applied models thus showed the following performance ranks in this step: Cu > RF > SVM > kNN.

According to Zeraatpisheh et al. (2019), although some models displayed a higher performance in the training dataset, lower performance in the validation dataset was observed [16]. The RF performance was poorer with RMSE = 5.15 cmole_c_ kg^−1^ and R^2^ = 0.41 in the validation subset, as compared to that in the training subset with RMSE = 2.74 cmole_c_ kg^−1^ and R^2^ = 0.86. Similarly, Zeraatpisheh et al. (2019), in estimating a calcium carbonate equivalent in central Iran, reported similar findings in regard to shifting the performance of RF models in calibration and validation datasets [16]. In general, both in training and validation datasets, RF was performed successfully to predict the soil CEC. Several scholars have reported the high capability of the RF model for predicting soil properties using auxiliary environmental variables [41,42]. While the model’s performance is dependent on several items, such as the target variable, number of field observations, input variables resolution, soil samples density, and auxiliary covariates type [43], it seems that in a similar environment with one hundred samples, the RF model could perform reliably. Figure 5 illustrates the relationships between observed and predicted soil CEC for the training as well as validation subsets. Excellent agreements were found in the training dataset; there was also a good agreement in the validation subsets for random forest (Figure 5a). In addition, the relationships between observed and predicted CEC values by the Cu model are presented in Figure 5b.

Overall, in the calibration step, the best model explained about 86% and 50% of the variability of CEC for the studied area in the validation stage. Although the RF model, as the nonlinear model might extract the nonlinear relationships between input and target variables, the evaluation of the capability of other nonlinear models, such as artificial neural network (ANN) or ensemble models, is suggested. On the other hand, several scholars have used various auxiliary variables to predict soil properties [44]. These variables presumably show the soil-forming factors controlling the variation observed in soil properties, such as soil CEC. Therefore, it seems that including other auxiliary covariates, such as legacy soil data and vis-NIR spectroscopy, could improve the model’s performance, which needs further investigation.

The best model (RF) was used for the preparation map of the soil CEC in the given studied area. Figure 6 illustrates the CEC distribution throughout the study area. However, the uncertainty map of prediction by the RF model is presented in Figure 6. The lowest CEC values were observed in the study area’s southwestern, north, and northeastern parts. The occurrence of the lowest CEC in the southwestern coincided with high altitude and mountainous areas with the lower clay and organic matter in the study area. While soils in the north and northeastern belong to Quaternary deposits, they are almost in the flood plain and have recent deposits with coarser particles and lower soil development. Lower soil evolution in these parts mainly led to the occurrence of clays with lower CEC, such as mica and chlorite [45]. However, the highest values were observed in the center towards the southern parts of the study area, which were dominantly covered by early Quaternary deposits. It seems, therefore, that soil CEC was enhanced in the early Quaternary deposits with a higher degree of soil development and the formation of smectite [46].

### 3.3. Variable Important Analysis

The results of the important variable analysis for two training and validation datasets are presented in Figure 7. As shown, the RF model was the best for the training dataset, and the Cu model was the same for the validation dataset. Valley depth derived from DEM was identified as the most important variable explaining the variability of CEC in the study area. The calculation of valley depth is according to the difference found between the elevation and a given interpolated ridge level. Ridge level interpolation applies the algorithm, which is conducted in the ‘Vertical Distance to Channel Network’ tool [47]. Valley depth, as the relative position of the valley, controls the variability of soil particles and soil organic matter influenced by soil erosion and deposition. Therefore, this topographic attribute seems to affect the soil CEC negatively; with increasing distance from the bottom of the valley, CEC is reduced. Significant and negative linear relationships were found between this topographic attribute and CEC (r = 0.53, *p* < 0.01) (Figure 7a). Following valley depth, elevation was recognized as the most significant variable. As mentioned in Section 3.2, the lower the CEC observed, the higher the elevated sites.

The slope significantly contributed to training and validation datasets in the latter ranking. Slope and topography could significantly contribute to the soil detachment variability in the landscape [48,49], and soil properties along the landscape are affected. On the higher slope, fine particles and soil organic matter are migrated from the surface soils and translocated to lower positions in the landscape. Therefore, lower CEC is expected in higher slopes, as confirmed by the results in Figure 8b. A positive relationship between clay content and CEC (R^2^ = 0.l569, *p* < 0.05) is presented in Figure 8c, indicating the effects of clay content on soil CEC. Furthermore, the translocation of fine clays with higher CEC, such as smectites derived from soil redistribution along the landscape and their accumulation in lower areas, may significantly contribute to increasing CEC. Moreover, in these parts of the area, cultivation activities and the application of manure promoted soil organic carbon. Soil organic carbon (SOC) has significant and well-known impacts on the enhancement of the cation exchange capacity of soils. A larger SOC concentration is commonly associated with a higher *CEC* [50]. A positive and significant relationship (R^2^ = 0.1024; *p* < 0.05) between SOC and CEC, as shown in Figure 8d, confirmed this speculation.

Among the remotely sensed indices, the ferric oxide index (SWIR1/NIR, see Table 1), ferrous mineral (SWIR1/NIR+Red), salinity index, and normalized difference moisture index (NDMI) were identified as the most influential variables. Considering the involved spectrum bands in the indices, most of them were associated with soil clay behavior, regarding the higher anions and cations and much more moisture content. Therefore, some scholars developed soil moisture indices derived from hyperspectral data to improve the estimation of soil clay and soil organic carbon [51,52,53]. However, further investigation is still required to explore the conceptual effects of CEC on these indices.

## 4. Conclusions

This study, as one of the first attempts to predict CEC in an arid as well as semiarid region, was performed to investigate the capability of four machine learning models and the use of some auxiliary environmental variables in the west of Iran. CEC values varied from 11.15 to 50.83 cmole_c_ kg^−1^ with a range of 39.68 cmole_c_ kg^−1^. The high variation of soil CEC in the study area could be mainly attributed to various contributions of different clays with different activities in diverse locations. According to the results of the X-ray analysis, clay types varied from low activity, such as illite and chlorite, to high ones, such as vermiculite and smectite. The modeling results also show that RF could be the best model in the training dataset by explaining about 86% of the variability of CEC in the given study area. On the contrary, the Cu model had the highest performance in the validation dataset.

Meanwhile, kNN had an inferior performance in both datasets. According to the variable importance analysis, valley depth, elevation, and slope from the topographic attributes could show the impacts of soil redistribution along the landscape as induced by soil erosion and deposition. Spatial distribution derived by the best machine learning model showed that the early Quaternary deposits in the study area center with high soil development degree had the highest CEC values. In contrast, shallow and undeveloped soils in the mountainous sites had the lowest values. For future studies, it is recommended to use other powerful machine learning models and additional auxiliary variables, such as oil legacy data and vis-NIR spectroscopic data, to improve the prediction accuracy.

## Figures and Tables

**Figure 1 sensors-22-06890-f001:**
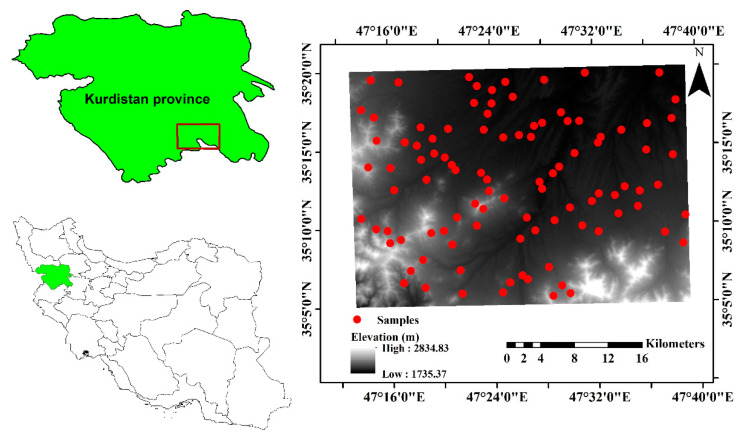
Location of the study area and spatial distribution of the studied points in Kurdistan Province, west of Iran.

**Figure 2 sensors-22-06890-f002:**
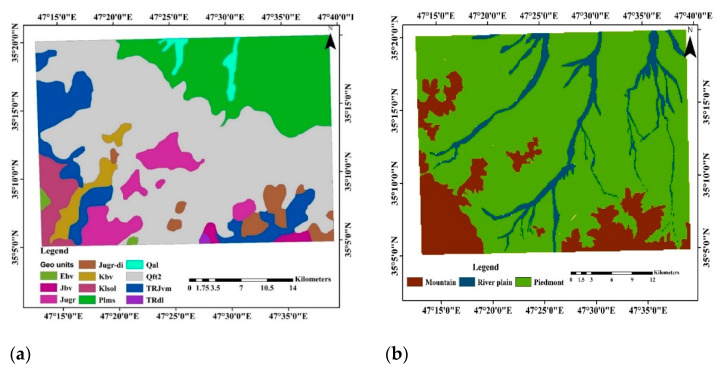
Illustration of the distribution of geologic (**a**) and geomorphic (**b**) units in the study area, west of Iran.

**Figure 3 sensors-22-06890-f003:**
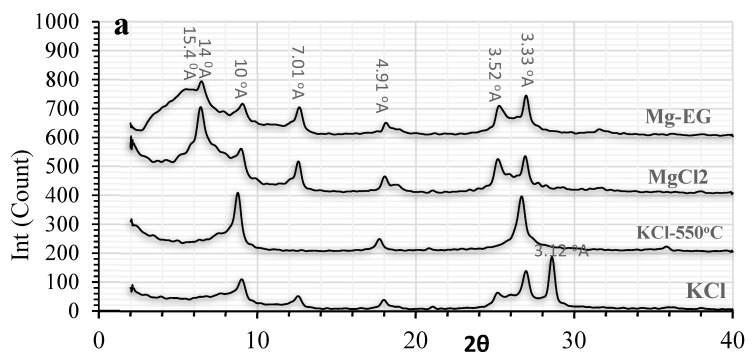
X-ray diffractograms of the clay fraction in three selected soils with various CEC content, (**a**) soil sample with CEC = 38.7 cmole_c_ kg^−1^, CA:2.85; (**b**) soil sample with CEC = 26.7 cmole_c_ kg^−1^ CA:1.51; (**c**) soil sample with CEC = 18.3 cmole_c_ kg^−1^, CA:1.02.

**Figure 4 sensors-22-06890-f004:**
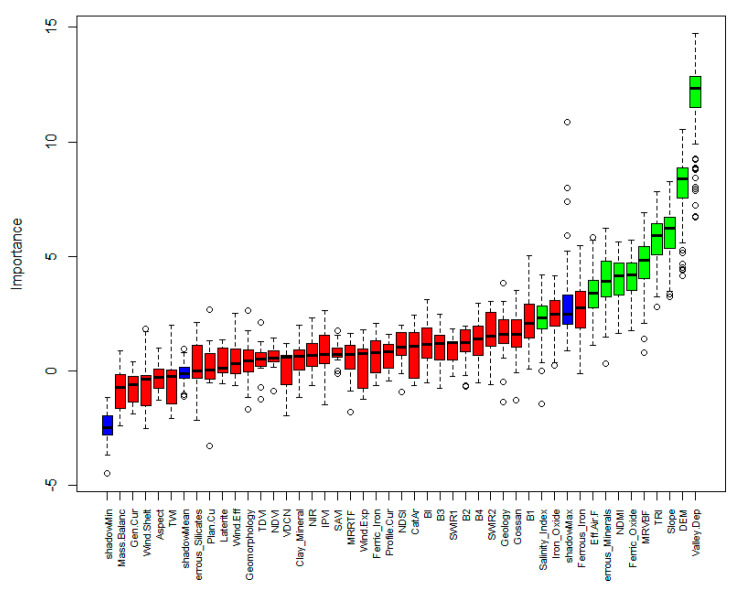
Brouta importance results for CEC in the study area in the west of Iran (Green and red box plots represent Z scores of confirmed and rejected attributes, respectively; blue box plots correspond to tentative attributes).

**Figure 5 sensors-22-06890-f005:**
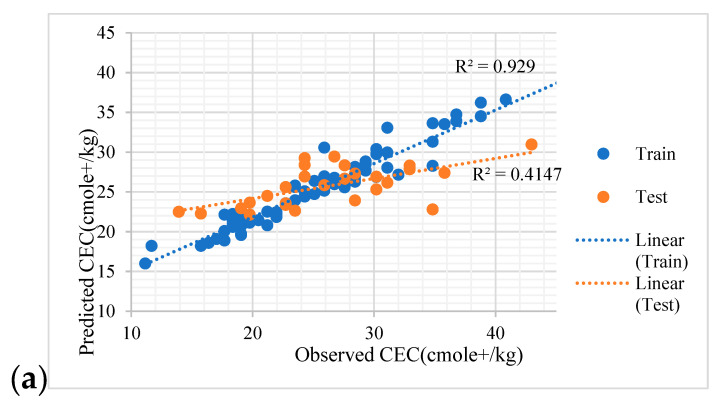
Observed CEC values versus the predicted values by RF model (**a**) and Cu model (**b**) in the training and testing datasets.

**Figure 6 sensors-22-06890-f006:**
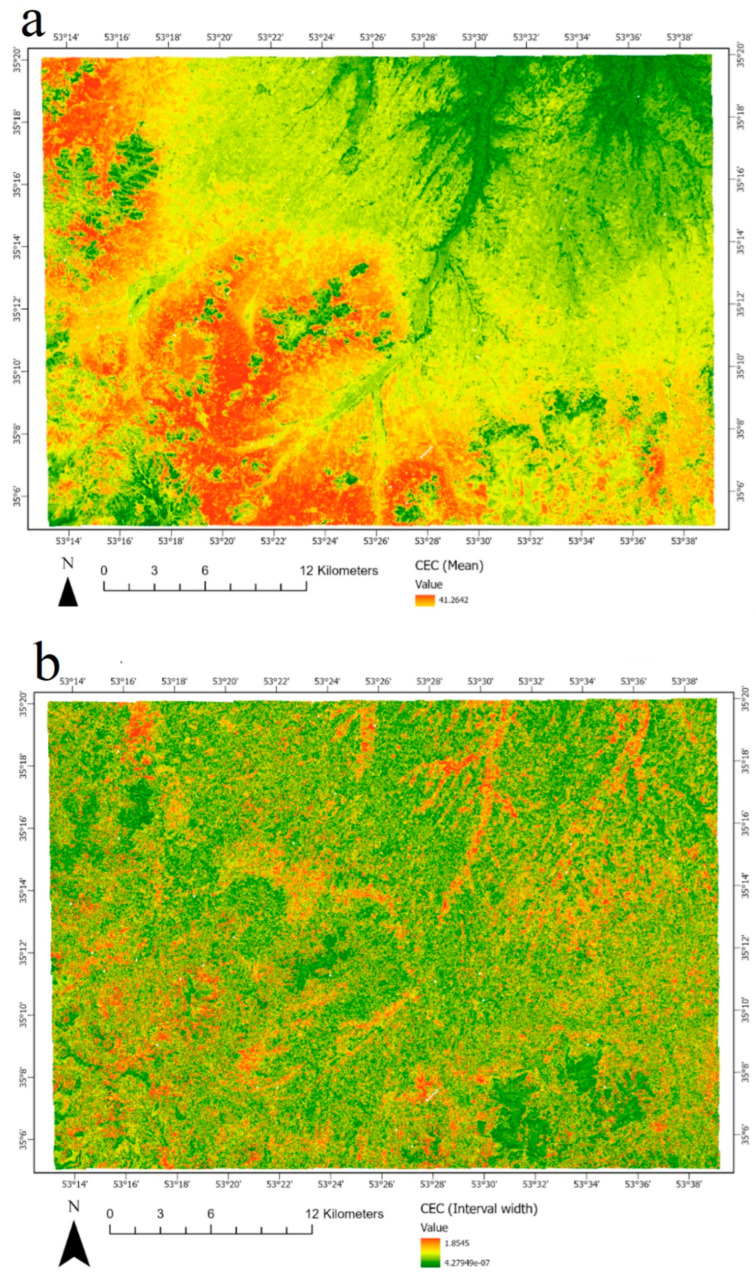
Predicted CEC (**a**) and associated uncertainty map (width of the 90% prediction interval) (**b**).

**Figure 7 sensors-22-06890-f007:**
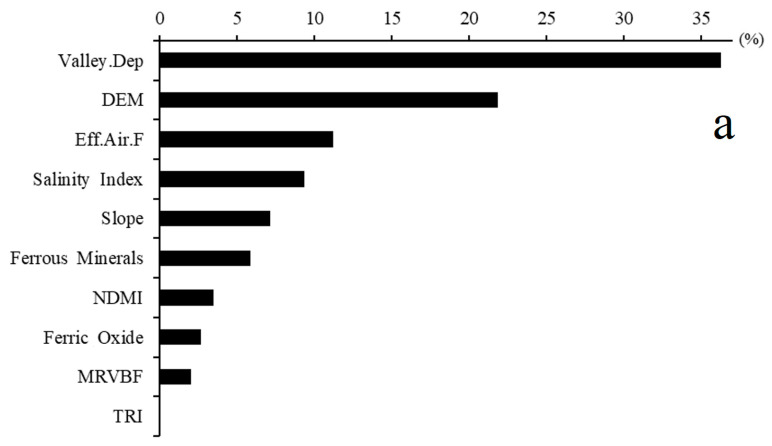
Importance variable analysis for predicting CEC by the best model for training, RF (**a**), and validation dataset, Cu (**b**); Valley. Dep: Valley depth; DEM: digital elevation model; TRI: Terrain ruggedness index; NDMI: Normalized difference moisture index; MRVBF: Multiresolution of ridge to flatness index, Eff.Aif.F: Effective airflow height.

**Figure 8 sensors-22-06890-f008:**
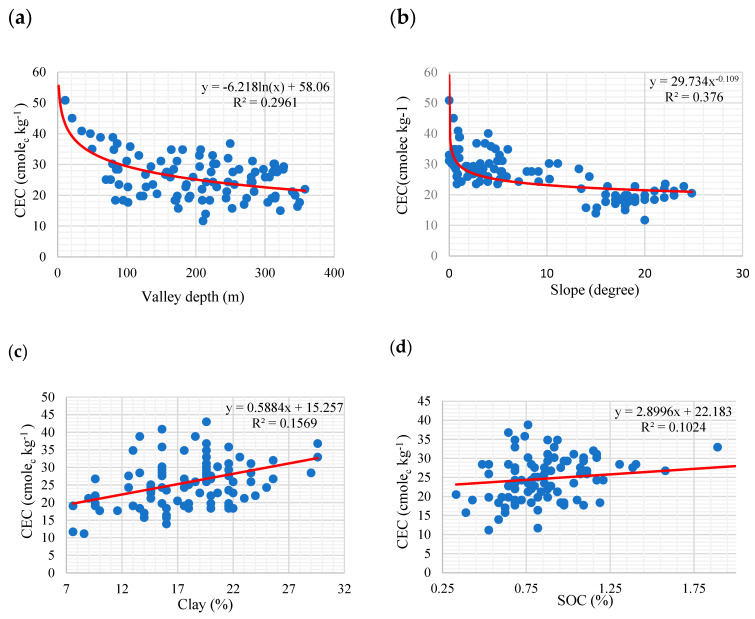
Relationships between some environmental variables and soil proprieties with CEC content in the study area. (**a**) valley depth vs. CEC, (**b**) slope vs. CEC, (**c**) clay vs. CEC, and (**d**) SOC vs. CEC.

**Table 1 sensors-22-06890-t001:** A list of environmental covariates used to predict CEC in the study area (Resolution = 30 m).

Covariates	Definition	Reference
Band 1 (0.433–0.45 µm)	Reflectance value of Landsat satellite band	Landsat satellite
Band 2 (0.45–0515 µm)	Reflectance value of Landsat satellite band (Blue)	Landsat satellite
Band 3 (0525–0.605 µm)	Reflectance value of Landsat satellite band (Green)	Landsat satellite
Band 4 (0.63–0.69 µm)	Reflectance value of Landsat satellite band (Red)	Landsat satellite
Near-infrared (0.75–090 µm)	Reflectance value of Landsat satellite band (NIR)	Landsat satellite
Shortwave infrared (1.55–1.75 µm)	Reflectance value of Landsat satellite band (SWIR)	Landsat satellite
Normalized difference vegetation index (NDVI)	(NIR − Green)/(NIR + Green)	Foody et al., 2001
Soil-adjusted vegetation index (SAVI)	((NIR − Red)/(NIR + Red + L)) × (1 + L)	Huete, 1988
Infrared percentage vegetation index (IPVI)	NIR/(NIR + Red)	Crippen, 1990
Transformed difference vegetation index (TDVI)	(NIR − Red)/(NIR + Red) + 0.5	Huete, 1988
Normalized difference moisture index (NDMI)	(NIR − SWIR)/(NIR + SWIR)	Wilson et al., 2002
Normalized difference snow index (NDSI)	(Red − NIR)/(Red + NIR)	Major et al., 1990
Brightness index (BI)	((Red × Red) + (NIR ×NIR))ˆ^0.5^	Khan et al., 2005
Salinity index	(Blue − Red)/(Blue + Red)	Douaoui et al., 2008
Clay Mineral	(SWIR1/SWIR2)	Douaoui et al., 2008
Ferrous Mineral	(SWIR1/NIR+Red)	Douaoui et al., 2008
Ferrous Iron	(SWIR2/NIR)+ (Green/Red)	Douaoui et al., 2008
Ferrous Silicates	(SWIR2/SWIR1)	Douaoui et al., 2008
Ferric Iron	(Red/Green)	Douaoui et al., 2008
Ferric Oxides	(SWIR1/NIR)	Douaoui et al., 2008
Iron Oxide	(Red/Blue)	Douaoui et al., 2008
Laterite	(SWIR1/SWIR2)	Douaoui et al., 2008
Gossan	(SWIR1/Red)	Douaoui et al., 2008
DEM (m)	Elevation	Conrad et al., 2015
Aspect (degree)	Aspect area	Conrad et al., 2015
CA	Catchment area	Conrad et al., 2015
GC	General curvature	Conrad et al., 2015
MrRTF	Multiresolution of ridge to flatness index	Conrad et al., 2015
MrVBF	Multiresolution valley bottom flatness index	Conrad et al., 2015
PlCu	Plan curvature	Conrad et al., 2015
PrCu	Profile curvature	Conrad et al., 2015
Slope gradient (degree)	Average gradient above flow path	Conrad et al., 2015
TWI	Topographic wetness index	Conrad et al., 2015
Valley depth (m)	Relative position of the valley	Conrad et al., 2015
Effective air flow heights (m)	Calculates effective air flow heights	Conrad et al., 2015
MBI	Mass balance index	Conrad et al., 2015
Terrain ruggedness index (TRI)	Measures terrain ruggedness	Conrad et al., 2015
Vertical distance to channel network (m)	Calculates the vertical distance to a channel network base level	Conrad et al., 2015
Wind effect	Dimensionless index indicating areas exposed to wind	Conrad et al., 2015
Wind exposition	Dimensionless index highlighting wind-exposed pixels	Conrad et al., 2015
Geology map	Representing the various geological features	-
Geomorphology map	Representing the various geomorphic units	-

**Table 2 sensors-22-06890-t002:** Descriptive statistics of some soil properties in the given study area (*N* = 97).

Variable	Unit	Min	Max	Mean	SD	CV (%)	Skewness	Kurtosis
SOC	%	0.33	6.63	0.96	0.66	68.75	6.59	54.59
CEC	cmole_c_ kg^−1^	11.15	50.83	25.75	7.02	27.26	0.60	0.76
pH	−log[H+]	6.72	7.78	7.32	0.22	3.00	−0.17	0.25
EC	dS m^−1^	0.09	0.69	0.23	0.10	43.47	2.84	9.20

SOC: Soil organic carbon; CEC: Cation exchange capacity; EC: Electrical conductivity; SD: Standard deviation; CV: Coefficient of variation.

**Table 3 sensors-22-06890-t003:** The best model parameters for predicting CEC in the study area.

Model	kNN	RF		SVM		Cu	
Parameter	k	ntree	mtry	sigma	C	committees	neighbors
Value	19	550	5	0. 3744	0.5	10	0

**Table 4 sensors-22-06890-t004:** Performance results of the applied models to predict CEC and clay activity in the study area (bold values indicate high model performance).

Model	Training							Validation						
	ME	RMSE	r^2^	R^2^	rhoC	RPD	RPIQ	ME	RMSE	r^2^	R^2^	rhoC	RPD	RPIQ
RF	**0.01**	**2.74**	**0.93**	**0.86**	**0.9**	**2.67**	**3.74**	−0.7	5.15	0.41	0.34	0.43	1.25	1.49
SVM	−0.52	4.47	0.67	0.62	0.73	1.64	2.3	−1.29	5.35	0.34	0.29	0.45	1.21	1.44
kNN	0.59	6.38	0.24	0.23	0.36	1.15	1.61	−0.49	5.72	0.19	0.18	0.32	1.13	1.34
Cu	−0.11	4.89	0.56	0.55	0.68	1.5	2.1	**−1.02**	**4.51**	**0.53**	**0.49**	**0.64**	**1.43**	**1.71**

The bold showed the highest performance for models.

## Data Availability

Data are contained in the tables of the article.

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
