# Peer review of "Comparison of Different Machine Learning Methods for Predicting Cation Exchange Capacity Using Environmental and Remote Sensing Data"

_sensors, 2022, doi:10.3390/s22186890_

Round 1

Reviewer 1 Report

This article discusses spatially incorporating environmental and remote sensing data to predict cation exchange capacity by applying machine learning approaches.

The authors present solid proof of concept, but they must explain how they obtained their graphs and describe their testing environment.

The authors should address the issues I have noted in their submitted article (Please refer to my comments on the document.)

Author Response

Response to Reviewer #1

General Comments

This article discusses spatially incorporating environmental and remote sensing data to predict cation exchange capacity by applying machine learning approaches. The authors present solid proof of concept, but they must explain how they obtained their graphs and describe their testing environment. The authors should address the issues I have noted in their submitted article.

Answer: Many thanks for the encouraging comments above. All grammatical corrections were made.  According to your valuable comments, we did almost of as well as we could, and the following considerations are presented for more clarifications:

Page 1, line 15: You need to distill your Abstract to make it shorter.

Answer: The abstract was shortened as well as we could.

Page 1, line 16: which behaviors?:

Answer: It was revised to: “fertility and mechanical behaviors” 

Page 1, line 33: You should not discuss values in your Abstract but rather discuss impact.

Answer: the values were removed, and just impacts were discussed.

Page 1: Line 40: You need to move this sentence to the end of your conclusion, as part as your future work.

Answer: The sentence was moved to the conclusion section

Page 3: Line 110: It?

Answer: It was corrected. 

Page  4 , Figure 1: You need to specify how did you obtain these graphs.

Answer: The graphs were prepared in ArcGIS and the distribution of points was displayed in The ArcGIS software based on the real data was recorded by GPS. This information for the paper regarding digital soil mapping is common.

Page 8, line 238: You need to describe your testing environment.

Answer: The testing environment already described in the methods and materials section, a part of Kurdistan province in the west of Iran, detailed of the study area, is provided in lines 108-118.

In tables 1 and 2: You need to specify how did you obtain these values.

Answer: these values are the parameters of models and validation criteria obtained by the R software that is normally used in digital soil mapping and already mentioned in method section line:  213. The validation criteria in Table 2 were calculated by the equations 2 to 5.

Figure 5: Why are you using these Chinese notations?

Answer: these labels are created after PDF made by the online submission.

According to Figures that you mentioned how they obtained:

Answer: Some figures are the outcome of R software for the spatial distribution of variables (Figure 6) and their importance variable (Figures 4 and  7). Figures 5 and 8 are the linear relationships of variables obtained by Excel software. Figure 3 is the out printing of X-ray diffraction analysis of the soil samples.  Figures 1 and 2 also show the thematic maps that were prepared in the ArcGIS by the authors.  

Reviewer 2 Report

Impression:

The authors try to use four machine learning methods to examine the capability of topographic features and remote sensing data to estimate the Cation exchange capacity (CEC) of the soil to minimize the laboratory cost in West Iran. The common agreement of soil CEC varies according to the clay %, the type of clay, soil pH, and the amount of organic matter. However, how to use the DEM, remote-sensing variables, and thematic maps to predict those four factors, especially the amount of organic matter, authors need to emphasize that. Otherwise, it will become just another GIGO machine learning practice.

Line-by-line comments:

The current title might not fit the three goals addressed at the end of the introduction. “Mapping” and “Spatial Predict” has different scope of work to cover. Suggests emphasizing the “comparison of different machine learning methods” and “determining the important variables” which can explain the CEC variability in the semi-arid region.

The abstract is too wordy; please be concise and informative.

Line 60: please put the name of citation #7 here, and then add the citation. For example: “as described by Rhoades [7],”

Figure 1: The current maps are too busy and small to see the detail information. Please refer to figure 1 of this paper https://www.tandfonline.com/doi/full/10.1080/00103624.2019.1604728 to simplify your figure 1.

Line 122: The typo of “S soil organic carbon…”

Table 1: typo near the “Calculates effective air flow heights”. Is the “Effective wwwwwww heights (m)” all correct? What kind of variables are extracted from the Geology and Geomorphology map also needs to be explained here or in a separate table.

Line 179: “CCE” or “CEC”?

Results and Discussion: again, the authors need to explain why they only focused on the clay type and percentage to estimate the CEC but ignored the soil pH and the amount of organic matter in this section.

Bottom of page 10: in the main text, authors sometimes used the “Cubist” and “Cu” to represent the same method. Please be consistent throughout the whole article.

Figure 5: Why is the Chinese notation on the sub-figure a and b?

Bottom of page 12: “……the best model explained about 86 % and 50 %of the variability of CEC for the studied area in the validation stage…….”. So, which one is the TRUE variability of CEC?

Figure 6: authors need to explain each sub-figures in the caption (and also in the main text). You cannot just leave it for the audience to guess what it is. (Especially, what is the “interval width”?)

Figure 7: If authors want to use “Eff.Air.F” to represent some variable due to the space issue, they need to explain it in the caption. Also, remove the extra underline from the variables’ name, for example, “Ferrous Minerals”

Figure 8: the x-axis tile of several sub-figures was blocked by the axis numbers. Please fix it. Also, the font color and size are different among the sub-figures.

Line 295-298: “SOC” needs to be defined first before use in the following sentence. Where is this SOC come from? The SOC is not in the list of figure 7 or anywhere across this article. The authors need to explain this clearly. If this SOC is from the field experience of 95 soil samples, the authors also need to list all the parameters they acquired from samples and used for the model calibration. Without including the SOC in the model calibration, you cannot say: “this factor is important to CEC, but I don’t use that in my model!” The sub-figure of SOC to CEC is in figure 8d, not figure8b. BTW, the R^2 for Clay% and SOC% are very small. How can you assert the relationship with CEC is significant?

Line 304-305: As mentioned the SOC is important; why don’t you include that in the analysis?

Suggestion:

There are many typos and inconsistencies across this article including the figures. Authors must also explain the rationales for using machine learning to estimate CEC across the study area. I suggest that the authors need to do a major revision before it can be published.

Author Response

Response to Reviewer  #2

Impression: The authors try to use four machine learning methods to examine the capability of topographic features and remote sensing data to estimate the Cation exchange capacity (CEC) of the soil to minimize the laboratory cost in West Iran. The common agreement of soil CEC varies according to the clay %, the type of clay, soil pH, and the amount of organic matter. However, how to use the DEM, remote-sensing variables, and thematic maps to predict those four factors, especially the amount of organic matter, authors need to emphasize that. Otherwise, it will become just another GIGO machine learning practice.

Answer: So many thanks for your valuables and precise considerations. As you know, the development of the pedotransfer functions has a long history in soil science to predict hard-measurable variables, these recent years using, machine learning models have been developed to do this more efficiently. For precise spatial management, we need continuous and acceptable maps with the minimum cost. You truly mentioned the factors affecting CEC, but as we believe that the soil forming factors (geology, topography, vegetation ….) control the intrinsic soil properties, we speculate that these variables could be used for prediction reasonably.  We also confirm several times in the manuscript that some important auxiliary variables have reasonable relationships with the factors influencing CEC.

 Line-by-line comments: The current title might not fit the three goals addressed at the end of the introduction. “Mapping” and “Spatial Predict” has different scope of work to cover. Suggests emphasizing the “comparison of different machine learning methods” and “determining the important variables” which can explainthe CEC variability in the semi-arid region.

Answer: The title was revised according to your suggestion and changed to “Comparison of different machine learning methods for predicting cation exchange capacity using environmental and remote sensing data”

The abstract is too wordy; please be concise and informative.

Answer: It was revised.

 Line 60: please put the name of citation #7 here, and then add the citation. For example: “as described by Rhoades [7]

Answer: It was corrected.

” Figure 1: The current maps are too busy and small to see the detail information. Please refer to figure1of this paper https://www.tandfonline.com/doi/full/10.1080/00103624.2019.1604728 to simplify your figure 1.

Answer: Figure 1 was revised according to your suggestion, and that reference was very relevant and included in the list of references in the revised manuscript.

Line 122: The typo of “S soil organic carbon…”

Answer: It was corrected.

 Table 1: typo near the “Calculates effective air flow heights”. Is the “Effective wwwwwwwheights (m)”all correct? What kind of variables are extracted from the Geology and Geomorphology map also needs to be explained here or in a separate table.

Answer: The typo was corrected. It was “Effective air flow heights (m)” .  In the geology and geomorphology map , just the categorical geomorphic units and lithologic units were included in the modeling, that is usual for thematic maps.

Line 179: “CCE” or “CEC”?

Answer: CEC is correct.

Results and Discussion: again, the authors need to explain why they only focused on the clay type and percentage to estimate the CEC but ignored the soil pH and the amount of organic matter in this section.

Answer: New table was included in the revised manuscript, the results showed the pH in the study area has no variation and has no significant effects on CEC, because of that, Calcium carbonate presence mediates the pH in the range of 6.72 to 7.78. Therefore the most effective variables are clay and clay type and SOM.

Bottom of page 10: in the main text, authors sometimes used the “Cubist” and “Cu” to represent the same method. Please be consistent throughout the whole article.

Answer: Your suggestion was considered.

Figure 5: Why is the Chinese notation on the sub-figure a and b?

Answer: It is a mistake produced by the Journal website.

Bottom of page 12: “……the best model explained about 86 % and 50 %of the variability of CEC for the studied area in the validation stage…….”. So, which one is the TRUE variability of CEC?

Answer: You mention to a good point, but in reality, the explained variability depended on the sample size and the location of the samples that were collected. There is no certain number for defining variability. If you change the dataset, the CV and other criteria certainly will change. There is the outcome for our dataset.

Figure 6: authors need to explain each sub-figures in the caption (and also in the main text). You cannot just leave it for the audience to guess what it is. (Especially, what is the “interval width”?)

Answer: The figure was revised and interval width was defined in the caption.

Figure 7: If authors want to use “Eff.Air.F” to represent some variable due to the space issue, they need to explain it in the caption. Also, remove the extra underline from the variables’ name, for example, “Ferrous Minerals”

Answer: Your suggestions were considered in the revised manuscript, and definitions of abbreviates were included in the caption.

Figure 8: the x-axis tile of several sub-figures was blocked by the axis numbers. Please fix it. Also, the font color and size are different among the sub-figures.

Answer: They were corrected.

Line 295-298: “SOC” needs to be defined first before use in the following sentence. Where is this SOCcome from? The SOC is not in the list of figure 7 or anywhere across this article. The authors need to explain this clearly. If this SOC is from the field experience of 95 soil samples, the authors also need to list all the parameters they acquired from samples and used for the model calibration. Without including the SOC in the model calibration, you cannot say: “this factor is important to CEC, but I don’t use that in my model!” The sub-figure of SOC to CEC is in figure 8d, not figure8b. BTW, the R^2 for Clay%andSOC% are very small. How can you assert the relationship with CEC is significant?

Answer:  SOC was defined in the sentence. We have measured the SOC in the research, but we did not intend to use it as covariate because of its measurement is cost and time consuming. But we used it just for clarification and explanation and cause – effects relationships between auxiliary variables and CEC in our discussion, anyway for more clarification we added a new table (Table 1), included some soil properties (SOC, CEC, pH….) and their statistics parameters.  Yes we did not use that in modeling. Regarding the R2 values, these values are R2 for the fitted line, but when they changed to “r” they are so higher and based on their numbers (N), the values were significant statistically.

Line 304-305: As mentioned the SOC is important; why don’t you include that in the analysis?

Answer: Inclusion of soil properties could be a scenario,. But we intended in this research to use easily available data.

Suggestion: There are many typos and inconsistencies across this article including the figures. Authors must alsoe xplain the rationales for using machine learning to estimate CEC across the study area. I suggest that the authors need to do a major revision before it can be published.

Answer: The manuscript was revised grammatically.  

Thanks again for all your contributions to improving the original manuscript.

Round 2

Reviewer 2 Report

Line 26-29: it seems the typo. 

Line 310-324: The description in this section is fine and supported by many references. However, the R-square (Coefficient of determination) in figure8 does not support your discussion here. Authors need to explain more clearly what kind of statistical methods they use to support the "positive and significant relationship (p<0.05)" (Line 323). As I previously mentioned, the SOC and CEC figure should be "Figure 8d" not "8b".

BTW, in the authors' replying "...Regarding the R2 values, these values are R2 for the fitted line, but when they changed to “r” they are so higher and based on their numbers (N), the values were significant statistically....". People usually use the R2 (R-square, Coefficient of determination) to represent the linear regression relationship. Not sure what the "r" means and where it comes from! If authors use methods other than the "regular linear regression analysis" (actually you shouldn't!), authors need to explain a clear method here and consider removing the regression equation and R2 value from the (figure8) charts and adding more relevant information. More definitions of R2 can be found online or here: https://en.wikipedia.org/wiki/Coefficient_of_determination.

Author Response

Response to Reviewer #2

Line 26-29: it seems the typo. 

Answer: It was corrected

Line 310-324: The description in this section is fine and supported by many references. However, the R-square (Coefficient of determination) in figure8 does not support your discussion here. Authors need to explain more clearly what kind of statistical methods they use to support the "positive and significant relationship (p<0.05)" (Line 323).

Answer: Thanks for your comments. According to R2 and “r” values for a pair of variables (X and Y), the results of R2 derived from simple linear regression, and r coefficient derived from the Pearson method, have an exactly similar concept, just the R2 is square of “r.” Therefore, using the table for evaluating the significance of the  “r” value or tables for evaluating the significance of the  R2 value, similar results are achieved based on the N=97.    [R2= 0.1024; r=0.32], whenever we use the significant relationship between two variables, we have already checked the corresponding statistical tables.   

As I previously mentioned, the SOC and CEC figure should be "Figure 8d" not "8b".

Answer: It was corrected.

BTW, in the authors' replying "...Regarding the R2 values, these values are R2 for the fitted line, but when they changed to “r” they are so higher and based on their numbers (N), the values were significant statistically....". People usually use the R2 (R-square, Coefficient of determination) to represent the linear regression relationship. Not sure what the "r" means and where it comes from! If authors use methods other than the "regular linear regression analysis" (actually you shouldn't!), authors need to explain a clear method here and consider removing the regression equation and R2 value from the (figure8) charts and adding more relevant information. More definitions of R2 can be found online or here: https://en.wikipedia.org/wiki/Coefficient_of_determination.

Answer: Please see my comment on the previous query. However, we add the R2 value in the parenthesis in the text in the new version of the manuscript.